# Functionalization of PET with Phosphazene Grafted Graphene Oxide for Synthesis, Flammability, and Mechanism

**DOI:** 10.3390/ma14061470

**Published:** 2021-03-17

**Authors:** Lifei Wei, Rui Wang, Zhiguo Zhu, Wenqing Wang, Hanguang Wu

**Affiliations:** 1Polymer Research Institute, Sichuan University, No. 24 South Section 1, Yihuan Road, Chengdu 610065, China; wlf13581518318@163.com; 2School of Material Science and Engineering, Beijing Institute of Fashion Technology, No. A2, East Yinghua Street, Chaoyang District, Beijing 100029, China; clyzzg@bift.edu.cn (Z.Z.); 20180021@bift.edu.cn (W.W.); 20180064@bift.edu.cn (H.W.)

**Keywords:** PET, graphene, flame retardant

## Abstract

Significant improvement in the fire resistance of polyethylene terephthalate (PET) while ensuring its mechanical properties is a tremendous challenge. A novel flame retardant (GO-HCCP, graphene oxide-hexachlorocyclotriphosphazene) was synthesized by nucleophilic substitution of the graphene oxide (GO) and hexachlorocyclotriphosphazene (HCCP) and then applied in PET by an in situ polymerization technique. The scanning electron microscope (SEM) showed a better dispersion of GO-HCCP than GO in the PET matrix. The char yield at 700 °C increased by 32.5% with the addition of GO-HCCP. Moreover, the peak heat release rate (pHRR), peak smoke produce rate (pSPR)and carbon monoxide production (COP)values significantly decreased by 26.0%, 16.7% and 37.5%, respectively, which indicates the outstanding fire and smoke suppression of GO-HCCP. In addition, the composites exhibited higher elastic modulus and tensile strength without compromising the toughness of PET matrix. These significantly reduced fire hazards properties are mainly attributed to the catalytic carbonation of HCCP and the barrier effect of GO. Thus, PET composites with good flame-retardant and mechanical properties were prepared, which provides a new strategy for further flame retardant PET preparation.

## 1. Introduction

Polyethylene terephthalate (PET) plays an important part in many fields, including the clothing industry, films and home furnishings, due to its outstanding properties, including outstanding impact strength, high flexibility, good wrinkle resistance, comfort while wearing, etc. [1,2,3,4]. Nevertheless, the high inflammability and the melting drip behavior restrict its further applications, especially in electrical engineering, transportation and aviation [5]. Therefore, enhancing the flame-retardant properties of PET is becoming a critical problem and has attracted great attention in recent decades [6,7]. Many reliable flame retardants, strategies and methods for suppressing fires have been developed by delaying the ignition or decreasing the release of heat and smoke [8,9,10,11,12,13]. Different types of fillers such as clay [14], silica [15,16,17] and metal oxide nanoparticles [18,19,20,21] have been incorporated into the PET matrix, which can achieve better flame-retardant, but the effect depends mainly on the type and the contents of the particles used most of which would destroy the mechanical properties. Wang [22,23,24] and his colleagues made great progress in anti-dropping and flame-retardant properties of PET by introducing phosphorus-containing ionic monomers into the PET backbone; these composites can achieve a limiting oxygen index (LOI) value of 36%, and UL-94 can also reach V-0 rating. However, a relatively large amount of these monomers were also needed to achieve efficient properties. Therefore, preparing flame-retardant PET while ensuring its mechanical properties remains a great challenge. Hexachlorocyclotriphosphazene (HCCP) is a typical hexatomic ring composed of P and N atoms connected by single and double bonds. This structure with high phosphorus and nitrogen content endows HCCP great potential in flame retardancy. In addition, the chlorine atoms can be easily substituted by alcohols, phenols and amines, which not only make HCCP a great starting material for functionalization, but also enhance the thermal and chemical stability of HCCP [25]. Therefore, cyclophosphazene derivatives have found applications in flame retardancy of both natural (e.g., cotton) and synthetic materials (epoxy resin [26], polyurethanes).

Fang [27] synthesized [9,10-Dihydro-9-oxa-10-phosphaphenanthrene-10-oxideo] triphosphazene (DOPO-TPN) from DOPO and HCCP, and then blended it into PET. After adding 5 wt% DOPO-TPN, the PET/DOPO-TPN composites reached 34% of LOI value and V-0 rating of UL-94. Wang [28] et al. prepared tubular one-dimensional poly (cyclotriphosphazene-melamine (PHMA) by the co-polymerization of hexachlorocyclotriphosphazene and melamine and also incorporated it into PET by blending; the PET-FR 5.0 (PET with 5 wt% PHMA) showed good flame retardancy and anti-dripping properties. However, HCCP is rarely used in PET by means of copolymerization because of its strong crosslinking with the precursor of PET, which may lead to the deteriorated mechanical properties of the obtained products. Hence, the cross-linking reaction with PET can be reduced by passivating one or two active functional groups of HCCP and then forming the linear polymer. Graphene oxide [29,30,31,32,33] is a two-dimensional macromolecule rich in oxygen-containing groups, and it can provide a good platform for further functionality. When graphene oxide (GO) grafted with HCCP was adopted as the flame retardant in PET, the break in the PET chain is believed to be suppressed due to the passivation of HCCP functional groups. On the other hand, the graft of HCCP on GO enlarges the interlayer spacing of GO, thus significantly preventing the aggregation and re-stacking of the GO layer. Zhang et al. [34] found that the addition of FGO (functionlized graphene oxide) grafted with HCCP and 3-aminnopropyltriethoxysilane into CE (cellulose plastics)could increase LOI and decrease peak heat release rate (pHRR) but also decrease the release of CO and CO_2_. Xu [35] also grafted HCCP and APTS (aminopropyltriethoxysiane) onto the surface of GO and then applied it in PSF. The pHRR and total heat release (THR) of the polysulfone (PSF) matrix with 5 wt% functional GO decreased by 49.20% and 14.87%. Herein, we firstly prepared a novel flame-retardant (GO-HCCP) through HCCP grafting onto GO sheets. Then, GO-HCCP was introduced to PET by in situ polymerization. In this study, the effects of GO-HCCP on flame retardant were investigated in terms of two main characteristics, i.e., both flame-retardant and mechanical properties. In addition, the mechanism for flame-retardant was also put forward.

## 2. Materials and Methods

### 2.1. Materials

Natural graphite powders (325 meshes) were obtained from Qingdao Huatai Lubricant sealing S&T Co. Ltd., Qingdao, China. Potassium permanganate (KMnO_4_, AR), concentrated sulfuric acid (H_2_SO_4_, 95–98%), tetrahydrofuran (THF) with analysis grades, triethylamine, ethanol, nitric acid (65–68%) and hydrochloric acid (37%) with chemical purity were all supplied by Sinopharmchemical Reagent Co., Ltd., Beijing, China. Hexachlorocyclotriphosphazene (HCCP) (≥99%) was obtained from Zibo Chemical Reagent Co., Ltd., Zibo, China. Terephthalic acids (PTA), Ethylene glycol (EG), antioxidant 1010 were received from Sinopec Tianjin Branch., Tianjin, China. Triphenyl phosphite was purchased from Chinese Pharmaceutical Company, Beijing, China, and antimony trioxide was supplied by Aladdin Reagent Shanghai Co., Ltd., Shanghai, China.

### 2.2. Synthesis of GO-HCCP

GO was prepared by the modified Hummers process, and then it was put into a flask containing THF of 300 mL, and the mixture was mixed under ultrasound for 30 min to obtain a uniform suspension. After adding triethylamine (0.3 mol, 30.3 g) into the flake, the flake was placed in a bath of ice water at 0–4 °C for 1 h. Then, THF (30 mL) dissolved with 6 g of HCCP was dropped into the flask within 2 h under stirring. The mixture was stirred for 2 h with nitrogen as a protective gas and finally heated at 60 °C for 3 h. Then, ethanol and THF were alternately cleaned several times to get a pure product. Finally, the product was dried in a vacuum oven at 60 °C for 12 h. The preparation route of GO-HCCP is shown in Figure 1.

### 2.3. Preparation of PET-GO-HCCP Composites

First of all, a certain amount of GO-HCCP was dispersed in the EG via vigorous stirring for 1 h and ultra-sonicating for 1 h. Then, 350 g PTA was added to the polymerizer, and 0.14 g antimony trioxide (as the catalyst) was also added. Then, the mixture was heated to about 220 °C under a nitrogen atmosphere, whereupon water was generated. After about 70 mL of water was removed, the transesterification step finished. Then, 15 g EG, 0.35 g 1010 (antihydrolytic agent) and 0.35 g triphenylphosphite were put into the reaction mixture for 30 min.

Then the temperature was increased to 270 °C, and a vacuum (<50 Pa) was provided. Polycondensation was then finished within 2~3 h once the mechanical stirring reached a specified torque. Neat PET, PET-0.4G, PET-0.2H composites were also synthesized with a similar method. The sample name and formula are listed in Table 3. The preparation route of GO-HCCP is shown in Figure 2.

### 2.4. Characterizations

Fourier transform infrared spectrometry (FTIR) was conducted by a Nicolet 6700 FTIR spectrometer (Thermo Fisher Scientific, Waltham, MA, USA) using the ATR method. The spectra were collected in the optical range of 400–4000 cm^−1^ with a scanning number of 64. The ATR crystal used to acquire the FTIR spectra was diamond. The X-ray diffraction patterns (XRD) of the samples were recorded by an X-ray diffractometer (Bruker D8, Bruker, Bremen, Germany) at a scanning rate of 0.1 per second in the 2θ range from 1° to 20° with Cu Kα tube and a Ni filter (λ = 0.154 nm), and the accelerating current and voltage were 20 mA and 40 kV, respectively. The X-ray photoelectron spectroscopy (XPS) was performed on an X-ray photoelectron spectrometer (Thermo ESCALAB 250, Thermo Fisher, Waltham, MA, USA) with Al Kα as excitation radiation (hv = 1486.6 eV) and the emission current and voltage were 5 mA and 10 kV, respectively. Raman spectroscopy was conducted by a SPEX-1403 laser Raman spectrometer (Renishaw inVia, London, UK) at an excitation wavelength of 532 nm. The morphology of samples and char residues was observed by JSM-7500F scanning electron microscope (JEOL, Tokyo, Japan) with an acceleration voltage of 10 kV. Thermogravimetric analysis (TGA) was conducted using a Thermal analyzer (TG 209, Netzsch, Selb, Germany), which uses nitrogen as a protective gas, and the gas flow rate was 100 mL min^−1^. The samples were kept within 5 to 10 mg and then heated from 30 °C to 700 °C at a ramp rate of 10 °C min^−1^. Differential scanning calorimetry (DSC, TA Q2000, TA Instruments, New Castle, DE, USA) was used to investigate the thermal transition behavior of composites. The samples were heated to 290 °C from 30 °C at a ramp rate of 10 °C min^−1^ for 1 min before it decreased from 290 °C to 50 °C at 10 °C min^−1^. Pyrolysis-gas chromatography–mass spectrometry (Py-GC-MS) experiments were performed by a pyrolyzer (EGA/PY-3030, Thermo Fisher, Waltham, MA, US), which was equipped with GCMS-QP2010 Ultra systems, and the carrier gas was helium. Real-time Fourier transforms infrared spectroscopy (RTFTIR) spectra were investigated using FTIR after the samples were heated in a muffle furnace for 1 min at different temperatures. The LOI tests were conducted on Dynisco LOI test instrument according to ASTM D 2863-97 standard. The specimen size used was 80 mm × 6.5 mm × 3 mm using Haak Mini Jet (Thermo Fisher, Waltham, MA, USA). The UL-94 vertical tests were carried out on a vertical burning instrument (CFZ-2,Nanjing Analytical Instrument Factory Co., Ltd., Nanjing, China) according to ASTM D3801 standard. The specimen size used was 130 mm × 13 mm × 3 mm. The cone calorimetry (Cone) test was conducted using a calorimeter (iCone, Fire Testing Technology Co., Ltd., East Grinstead, UK) according to the ISO 5660-1 standard under a heat flux of 50 kW m^−2^, and the sample with the dimensions of 100 mm × 100 mm × 3 mm was wrapped with aluminum foil. All the tests were repeated three times, and the results were averaged. The mechanical performance was measured by Instron electronic universal material strength tester (4302, Instron Corporation, High Wycombe, UK) at a speed of 10 mm·min^−1^ according to the ASTM D-638 standard. The width of the specimen was 4.0 ± 0.1 mm and the thickness was 2.0 ± 0.1 mm. All the samples were tested five times with mean values as the final data, with variance of plus or minus 5%.

## 3. Results

### 3.1. The Molecular Structure of GO-HCCP

The flame-retardant GO-HCCP was synthesized, followed by the identification of the grafting between GO and HCCP. Figure 3a shows the peaks at 1707, 1224, 1045 and 1363 cm^−1^ are ascribed to the C=O, epoxide group and the C-C skeleton of GO. For the FTIR absorption peaks of GO, HCCP and GO-HCCP, the broadband at 1628 and 3357 cm^−1^ correspond to the bending vibrations and stretching of O-H on the GO surface. The bands in the range 2500–2800 cm^−1^ corresponded to triethyamine hydrochloride, which indicates that the products should be further purified. For HCCP, the band at 593, 1180 and 873 cm^−1^ correspond to P-Cl, P-N and P=N bonds, respectively. Compared to GO and HCCP, the new absorptions band at 1032 and 835 cm^−1^ appeared due to stretch vibration of P-O-C in GO-HCCP, which indicated that GO had been successfully grafted by HCCP. The interaction between GO and HCCP was also confirmed by XRD (Figure 3b). A sharp diffraction peak was observed at 11° of GO, which corresponded to the (002) reflection of GO. Nevertheless, the (002) peak intensity decreased after the chemical modification of GO (GO-HCCP), and the (002) diffraction peak of GO shifted to 9.9°. According to 2dsinθ=nλ, the layer spacing of GO increased to 0.99 nm from 0.8 nm. This indicates that the organic molecules were grafted onto the surface of GO by nucleophilic substitution reaction, which led to the breaking of the stacking structure of GO sheets, leading to the enlarged interlayer spacing of GO via nucleophilic substitution reaction of HCCP onto GO. XPS characterization results of GO and GO-HCCP are displayed in Figure 3c. It is obvious that GO-HCCP exhibited a new adsorption peak at 400, 190.9 and 134.7 eV, which were attributed to the N1s, P2s and P2p derived from HCCP. The nanostructure was also confirmed by Raman analysis. Figure 3d presents the Raman spectra of GO and GO-HCCP. They all exhibited a typical broad D-band and G-band, with the D-band at 1335 cm^−1^ and G bands at 1593 cm^−1^ attributable to the defect/disorder-induced mode and graphene in-plane vibration, respectively [36]. The relative intensity ratio of D-band to G-band (I_D_/I_G_) is an important value to monitor the purity and functionalization of GO. In this study, the values of I_D_/I_G_ were 1.19 and 1.29 for GO and GO-HCCP, respectively. The increased I_D_/I_G_ value indicated that the introduction of sp^3^ defects disordered the graphene structure. Based on the aforementioned discussion, it is reasonable to state that functionalization of flame-retard GO-HCCP has been successfully synthesized with strong GO-HCCP chemical bonds. According to the decreasing mass in TGA, the grafted mass was approximately 26.62% compared with the mass of GO.

The SEM and energy-dispersive X-ray spectroscopy (EDS) image were showed in Figure 4. As we can see from the image, the surface of GO-HCCP contains a lot of C, O and P. At the same time, P element is homogeneously dispersed on the surface of lamellae, indicating the evenly grafting of HCCP.

### 3.2. The Characteristics of PET Composites

#### 3.2.1. Morphology

In order to evaluate the morphology and dispersion of GO-HCCP, SEM images were provided to reveal the dispersion information of composites at the micrometer scale. As shown in Figure 5, the surface of pure PET was quite smooth except for a few fragments while the surface of PET-0.4GO was relatively rough with some irregular protuberances from graphene, indicating worse distribution in the PET matrix. In contrast, it was found that the GO-HCCP hybrids were uniformly embedded into the PET matrix, indicative of good dispersion. The good dispersion of GO-0.4G-0.2H in the PET matrix could be attributed to the increase in layer spacing of the GO sheets, which was conducive to the exfoliation and more uniform dispersion of GO.

#### 3.2.2. Thermal Properties

The crystallization and thermal properties of PET and its composites were studied by DSC. The data and curve of second heating and cooling are presented Table 1 and Appendix A after erasing thermal history at 290 °C for 1 min [37]. Pure PET appeared at a single glass transition temperature (T_g_) of 71.1 °C; however, the values decreased to 49.4 °C and 51.4 °C with the incorporation of 0.4 wt% GO or 0.2 wt% HCCP, respectively. The lower T_g_ values were consistent with the reported values in the literature, which may be due to the interplay (interaction) between GO (HCCP) and EG (PTA) via multi-active group reactions. Furthermore, the crosslinked structure in PET-0.2G and PET-0.4H inhibits the formation of long chains and lowers the mobility of polymer chains [38]. As a result, both cold crystallization temperature (T_cc_) and melting temperature (T_m_) decreased to lower values in PET composites. Similarly, the melting crystallization temperature (T_mc_) shifted to lower values in the cooling process. By contrast, the observed T_g_, T_mc_ and T_m_ values of PET-0.4G-0.2H were higher than single composites. This was attributed to the mechanism of chain growth change to a routine (normal) two-dimensional linear rather than a three-dimensional network. This observation will be further discussed in the following part.

To further assess the thermal properties of the PET composites, TGA tests were used to characterize the thermal degradation and thermal oxygen degradation behaviors of PET and its composites, and the corresponding thermal decomposition parameters are summarized in Table 2 and Appendix A. The thermal degradation process of pure PET presents one step at 428.7 °C with a maximum loss weight rate, and the char residue was 8.3 wt% at 700 °C, which is due to the release of CO, CO_2_, H_2_O, acetaldehyde and oligomers with carboxyl and hydroxyl terminal groups. With the incorporation of GO into the PET matrix, the temperature at 5 wt% weight loss (T_5 wt%_) of PET-0.4G increased to 383.8 °C from 381.9 °C of PET, and this can be explained by the fact that graphene has a good barrier effect and can insulate heat transfer into the matrix. Therefore, the final residue of PET-0.4G increased to 8.8 wt% at 700 °C. After the incorporation of HCCP into PET composites, the T_5 wt%_ and the maximum mass loss rate temperature (T_max_) for PET-0.2H reached 391.8 °C and 435.6 °C, respectively. The temperature at 30 wt% weight loss (T_30 wt%_) and the temperature at 50 wt% weight loss (T_50 wt%_) also increased by approximately 7 °C. The result demonstrated that the addition of HCCP improved the thermal stability and further delayed the decomposition of PET, which were related to the cyclic-type conjugated structure containing P and N in HCCP. After the incorporation of GO-HCCP into the PET matrix, the T_5 wt%_ for PET-0.4G-0.2H composite increased to 391.7 °C and the char residue at 700 °C (C_w_) increased to 11.0%, which increased by 32.5% compared with neat PET. The addition of GO-HCCP led to the compacted and insulated char layer, which could defend the inner PET matrix from further burning. Thermal oxygen stability of PET-GO-HCCP was also enhanced, as can be seen from the TGA curve of PET in the air atmosphere in Appendix A.

#### 3.2.3. Flammability

The LOI value and UL-94 of PET and its composites were tested, and the data are presented in Table 3. As we can see, the LOI value of pure PET is just 22%, which indicates that PET can easily ignite and burn in air, while the LOI was not improved by adding GO alone. PET-0.2H has a bit higher LOI than PET because of the addition of HCCP. For PET-0.4G-0.2H, its LOI was 24%. As for UL-94, all the samples had a V-2 rating.

The cone calorimeter is the most realistic method to evaluate combustion properties of polymers under a forced-flaming fire scenario. The pHRR, THR and the peak smoke production (pSPR) data are shown in Table 4 and Appendix A. The time to ignition (TTI) of PET-0.4G and PET-0.2H evidently decreased to 50 s and 57 s from 63 s for neat PET due to the good thermal conductivity of graphene and the early degradation of HCCP, while the TTI of PET-0.4G-0.2H increased to 58 s which was delayed by 8 s compared to pure GO addition. This can be attributed to the small amount of GO-HCCP, which improves the thermal stability of composites and retards the ignition of PET.

After ignition, PET burned fiercely, There was only one intensive heat release rate (HRR) peak that appeared with a pHRR of 875.9 kW·m^−2^ at 102 s, and the total heat release (THR) of PET was 60.7 MJ·m^−2^. With the incorporation of 0.4 wt% GO, the pHRR and THR increases to 916.0 kW·m^−2^ and 71.4 MJ·m^−2^ respectively, which is related to the good thermal conductivity of GO, meaning that heat can be transferred to the PET interior faster. Consequently, PET can burn more fully, and the CO_2_ production (CO_2_P) increased to 0.65 g/s from 0.62 g/s. For the sample of PET-0.2H, the heat and smoke release all showed the lowest values. This could be related to the earlier decomposing of HCCP, which may have released free radicals to capture flammable free radicals and then to break off the radical action [26]. The HCCP can also generate phosphate and pyrophosphate covered on the matrix surface to prevent the PET from further combustion. After 0.4G-0.2H was added, the pHRR, peak of smoke production rate (pSPR) and CO production (COP) of PET-0.4G-0.2H decreased by 26.0%, 16.7% and 37.5%, respectively, compared with neat PET. This is due to the barrier effect of graphene and the formation of P-O-C structure, which can act as the skeleton of carbon and promote carbonization.

In order to further study the flame-retardant mechanism, the results were quantified according to Equations (1)–(3), and the summary results are shown in Table 5.
E _Barrier_ = 1 − (^1^pHRR_FRPET_/^1^pHRR_PET_)/(^2^THR_FRPET_/^2^THR_PET_)(1)
E _Flame inhibition_ = 1 − ^3^MEHC_FRPET_/^3^MEHC_PET_(2)
E _Charring effect_ = 1 − ^4^TML_FRPET_/^4^TML_PET_(3)
where ^1^ pHRR and ^2^ THR are the peak of heat release rate and the total heat release, respectively; ^3^ MEHC is the mean effective heat of combustion; and ^4^ TML is the total mass loss.

As shown in Table 5, the proportion of barrier and charring effects of the PET-0.4G were estimated to be 11.1% and 4.4%, respectively, which indicates that the GO lamellar structure can play a certain shielding effect and char effect in the PET matrix. However, due to its good thermal conductivity, it would be hard to restrain the further combustion of the matrix, so the proportion of flame inhibition effect was just 0%. As for PET-0.2H, the proportion of barrier effect and charring effect increased to 12.6% and 9.5% due to the presence of P-O-C structure. The P-O-C structure can enhance the strength of the char layer and further enhance the barrier properties. After the 0.4G-0.2H was added, the barrier effect was further strengthened, reaching 29.1%, while the flame inhibition and charring effects showed a slight decrease, which is due to the antagonism between the well thermal conductivity of GO and flame-retardant of HCCP.

#### 3.2.4. Mechanical Properties

Figure 6 shows the tensile properties of all samples. The elastic modulus and tensile strength for PET are 834.6 MPa and 53.2 MPa. Compared to neat PET, the introduction of GO in PET led to the elastic modulus and tensile strength increasing by 13.25% and 27.22%, respectively. In contrast, the tensile strength decreased to 27.9 MPa after the HCCP was added into the PET matrix, which is due to the trifunctional structure of HCCP and the fact that it is easy to form a cross-linking structure. With the addition of GO-HCCP, both the elastic modulus and tensile strength increased by 43.25% and 4.12%, respectively. This is mainly due to the uniform dispersion of GO-HCCP hybrids and efficient transfer between the graphene nanosheets and PET matrix. 

### 3.3. Mechanism of Flame Retardation

To further investigate the flame retardation mechanism, the microstructures of the residual char were observed by SEM. Figure 7 shows the SEM image for char residue of PET and its composites at amplification power of 100 × (100 μm) and 500 × (10 μm), respectively. It is clearly seen that pure PET formed a char residue structure with cracks and many large holes, indicating a low effective barrier layer. When GO or HCCP are separately added to PET, the densities of the hole are slightly lower compared to that of pure PET. However, there are still big cracks in the char layer due to the release of gas, which make it hard to isolate the heat and mass transfer during combustion. In the case of PET-0.4G-0.2H, a swelling and continuous char was obtained on the surface of PET-0.4G-0.2H, and there was a higher char density with smaller holes. This is due to the combined effect of the graphene barrier and the P-O-C structure in the char layer.

Real-time Fourier transforms infrared spectroscopy (RTFTIR) was employed to reveal the evolution of chemical structures during thermally oxidative degradation of materials. Figure 8 depicts the spectra of PET and its composites at different temperatures. There are two peaks at 1451 and 2963 cm^−1^ corresponding to bending vibrations and stretching vibrations of -CH_2_- in the aliphatic chain for pure PET. The peaks at 1239 and 1712 cm^−1^ were assigned to the stretching vibrations of C-O-C and C=O, respectively. Moreover, the peak located at 1408, 871 and 721 cm^−1^ belongs to the skeleton, otho-position H and bending vibrations of benzene. When the pyrolysis temperature exceeds 400 ℃, the intensity of the peak drops sharply due to the decomposition of unstable aliphatic chain and C-O-C bonds. However, it is worth noting that there is still obvious absorption such as 1597, 818 and 698 cm^−1^ at 600 °C, implying the presence of many benzene and double bonds in the char layer. Compared with pure PET, the peak at 1174 cm^−1^ assigned to the P-O-C vibration in PET-0.4G-0.2H indicates the reaction of EG and GO-HCCP with high thermal stability. Therefore, the char layer composed of phosphorous-carbon complexes can be used as an effective barrier to the decomposition of the protective matrix at high temperatures [39].

As we can see in Figure 9a, for neat PET, the intensity ratio I_D_/I_G_ of the residue char was about 1.19. The incorporation of GO increases the value of I_D_/I_G_ to 1.29, which is due to the good thermal conductivity of GO. However, the presence of GO-HCCP reduces the I_D_/I_G_ values of the char to 1.14, which indicates the increases in graphitized carbon in the residue char. Hence, GO-HCCP enhances the char layers density and strength and shields the mass and heat transfer during the combustion process.

To further explore the structure of char residue, XPS was performed to investigate the composition of char residue of samples. Figure 9b shows the XPS spectra of the char residue of neat PET and its composites. In the wide spectra, four spectra show sharp peaks at 285.8 and 533.5 eV, corresponding to C and O elements, respectively. However, PET-0.2H and PET-0.4G-0.2H had the additional P2s peak at 134.7 and P2p peak at 190.9, suggesting the existence of phosphorus-based derivatives on the surface of residue char. The existence of phosphorus-based derivatives not only significantly increases the heat resistance of the materials, but also enhances the char-forming ability, so the PET-0.4G-0.2H is highly flame-retardant.

To study the thermal degradation process, Py-GC-MS tests were carried out to monitor the pyrolysis behaviors of PET and PET-0.4G-0.2H. The resulting data are shown in Table 6 and Figure 10 and Figure 11. The main pyrolysis products of neat PET are 34.1% Benzoic acid, 8.7% Terephthalic acid, 8.5% Methyl benzoate, 7.6% benzene, 7.2% Bibenzene, 6.3%p-ethylbenzoic acid, 5.1% Styrene, 5.0% p-Acetylacetophenone and 3.9% Acetophenone [40]. Based on the results, the proposed pyrolysis process of PET is shown in Figure 10. The PET generates carboxylic acid and olefinic end groups through a classical ester scission reaction were further transformed by ester scission, rearrangement and radical reactions.

Compared with neat PET, the pyrolysis products after the addition of the 0.4G-0.2H represents some differences in the relative concentrations and composition. The proposed pyrolysis process for PET-0.4G-0.2H is shown in Figure 11. The main pyrolysis products were 39.6% Benzoic acid, 11.3% Terephthalic acid 7.2% p-Acetylacetophenone, 5.2% benzene, 4.8% Methyl benzoate, 4.6% Terephthalic acid, 4.3% Bibenzene, 3.7% Styrene, 3.1% Acetophenone. The contents of benzene acid increased obviously, which indicates that the degradation of PET was inhibited, while the contents of benzene showed a significant decrease, which demonstrated the GO-HCCP could restrain the further degradation of benzene acid. The decline of styrene and methyl benzoate indicated that the deep degradation of olefinic end group was restrained, which can explain the improvement in thermal stability of the PET-0.4G-0.2H. Moreover, the decrease in biphenyl and terphenyl indicates that the presence of GO-HCCP could slow down the degradation process of PET.

Besides the pyrolysis product mentioned above, about 5.6% of newly formed pyrolysis products were formed at the retention time from 17 to 21 min, including 1, 2, 4-benzenetricarboxylic acid, 3, 5-dinitro-, trimethyl ester, Phthalazine-1,4(2H,3H)-dione, 2-(2-methyl-5-nitrophenyl), 5-Ethyl-5-phenylhydantoin, 8, 10, 18, 20, 21, 23-hexaene-2, 7, 12, 17-tetrone and o-(4, 6-Diphenyl-1, 3, 5-triazin-2-yl)phenol. These products were mainly amino group and triazine radicals generated by the decomposition of GO-HCCP generated, which reacted with phenyl and carboxylic acid generated by the pyrolysis reaction of PET to generate an orthophenanthroline ring, ammonia acid, nitriles, and nitrogen heterocyclic ring [41]. These products played an important part in suppressing deacidification and free radical reactions and thus catalyzing the formation of the char layer.

Combining the results of flammability measurements, the morphology of the char and TGA, the possible flame-retardant mechanism shown in Figure 12 is proposed as follows: (i) the tortuous path effect and barrier effect of GO can insulate the permeation of oxygen and volatile flammable gases, thereby decreasing the release of heat and smoke; (ii) the diluting and quenching effect of HCCP can reduce the concentration of oxygen and flammable gases, which can restrain combustion in gas phase effectively; (iii) phosphorus-based derivatives formed in condensed phase can significantly promote the formation of protective layer, preventing composites from the radiant heat and direct contact with flame. In addition, the grafted GO can homogenize the dispersion, which is of great value and significance for the further reduction of pHRR, pSPR and COP of PET.

## 4. Conclusions

In summary, GO was functionalized with HCCP and then introduced into PET to obtain the fabrication of PET composites. The successful graft of GO was verified by FTIR, XRD, XPS and Raman. The grafted GO showed uniform distribution in the PET. PET with GO-HCCP had a higher char residue of 13.2% at 700 °C compared to neat PET (8.3%). In cone calorimetry test, the PET-0.4G-0.2H exhibited good flame-retardant properties. Its pHRR, pSPR and COP decreased by 26.0%, 16.7% and 37.5% compared with pure PET. The barrier effect and tortuous path effect of GO and phosphorus-based derivatives generated by HCCP impede the release of smoke and combustible volatile during combustion and then preserved the interior of the matrix. Moreover, the PO· and PO_2_· released by HCCP can quench the flammable free radical such as H· and HO· and then restrain combustion in the gas phase. In addition, the composites exhibited higher elastic modulus and tensile strength without compromising the toughness of the PET matrix. As a result, the flame-retardant and mechanical properties of PET-0.4G-0.2H composites were simultaneously improved, caused by the incorporation of a small amount of GO-HCCP.

## Figures and Tables

**Figure 1 materials-14-01470-f001:**
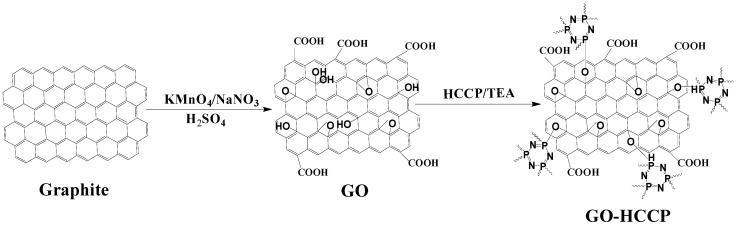
The preparation route of GO-HCCP (graphene oxide-hexachlorocyclotriphosphazene).

**Figure 2 materials-14-01470-f002:**
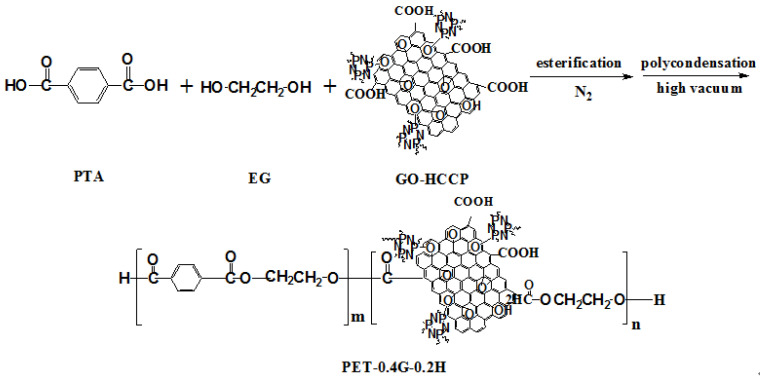
The preparation route of PET-0.4G-0.2H.

**Figure 3 materials-14-01470-f003:**
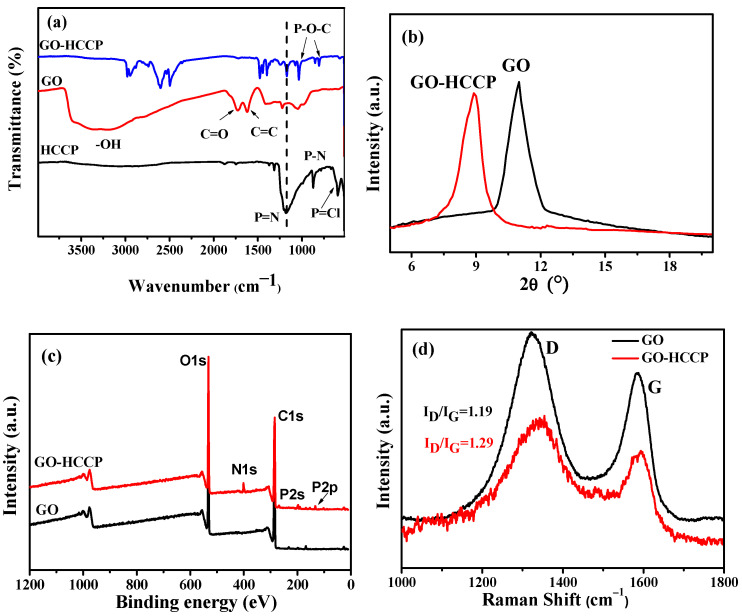
Molecular structure of the GO-HCCP. (**a**) FTIR (fourier transform infrared spectrometry), (**b**) XRD (X-ray diffraction), (**c**) XPS (X-ray photoelectron spectroscopy), (**d**) Raman, (**e**) TGA (thermogravimetric analysis) curve of GO, HCCP and GO-HCCP.

**Figure 4 materials-14-01470-f004:**
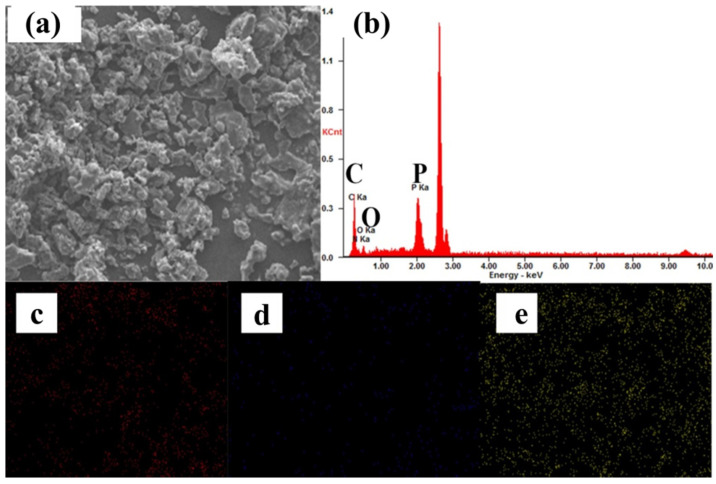
SEM-EDS of GO-HCCP.(**a**) SEM; (**b**) EDS; (**c**) C; (**d**) O; (**e**) P.

**Figure 5 materials-14-01470-f005:**
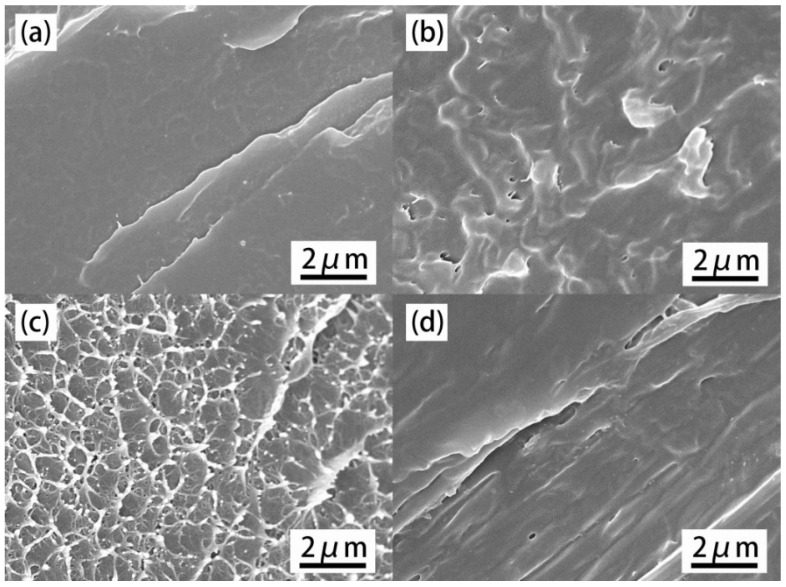
SEM images of neat PET and its composites’ fracture surfaces. (**a**) PET, (**b**) PET-0.4G, (**c**) PET-0.2H, (**d**) PET-0.4G-0.2H.

**Figure 6 materials-14-01470-f006:**
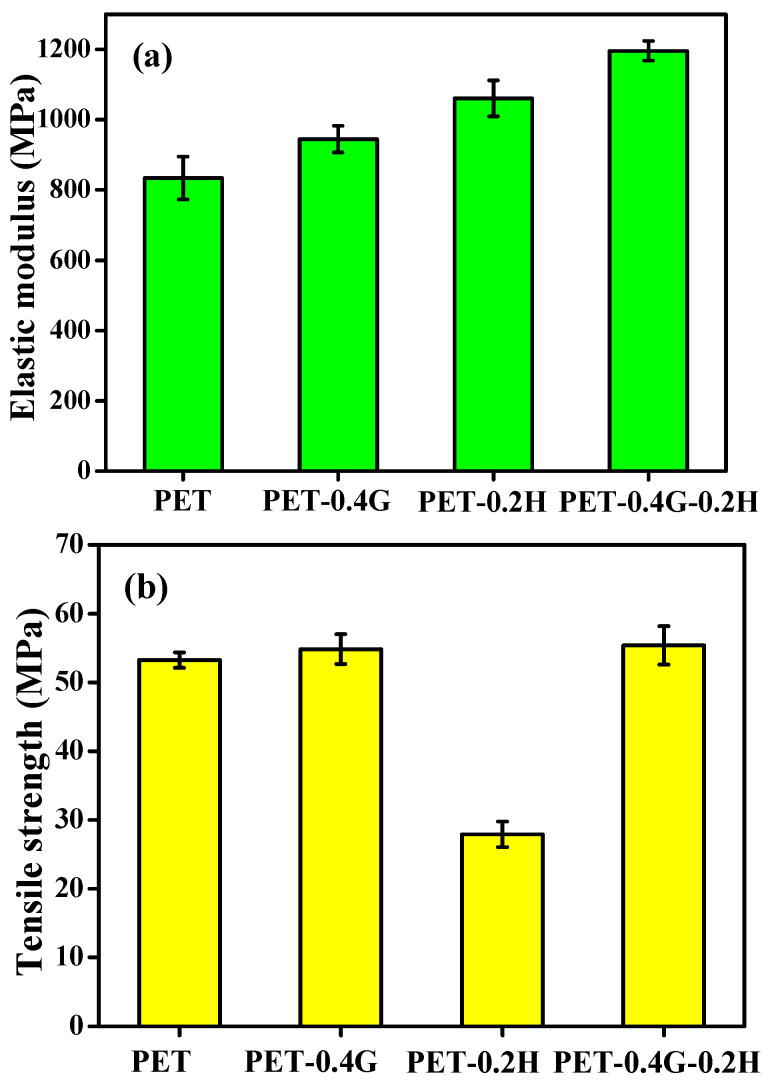
Mechanical properties of sample: (**a**) Elastic modulus, (**b**) Tensile strength

**Figure 7 materials-14-01470-f007:**
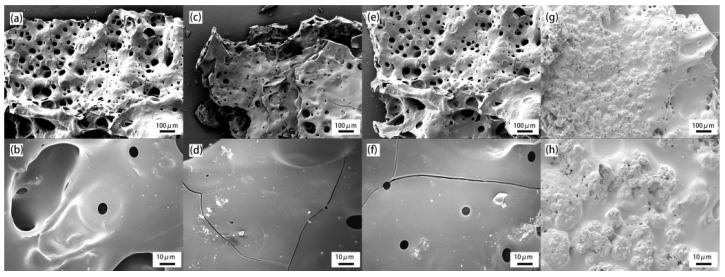
SEM images of chars for composites after the cone calorimetry tests. (**a**,**b**) PET, (**c**,**d**) PET-0.4G, (**e**,**f**) PET-0.2H, (**g**,**h**) PET-0.4G-0.2H.

**Figure 8 materials-14-01470-f008:**
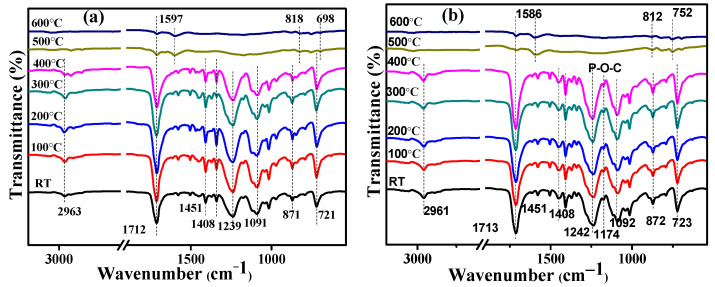
Real-time Fourier transform infrared spectroscopy (RTFTIR) spectra for condensed products of (**a**) PET (**b**) PET-0.4G-0.2H.

**Figure 9 materials-14-01470-f009:**
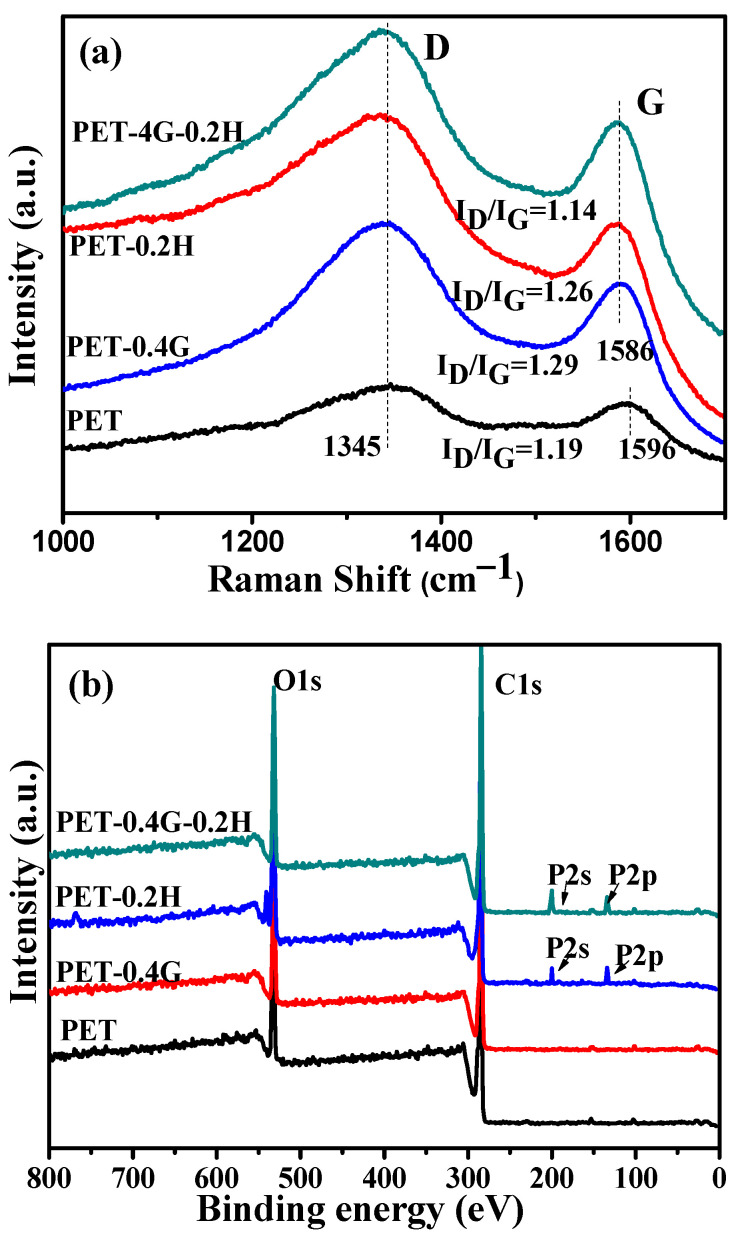
Raman spectra (**a**) and XPS (**b**) scan spectra of char residue of neat PET and its composites.

**Figure 10 materials-14-01470-f010:**
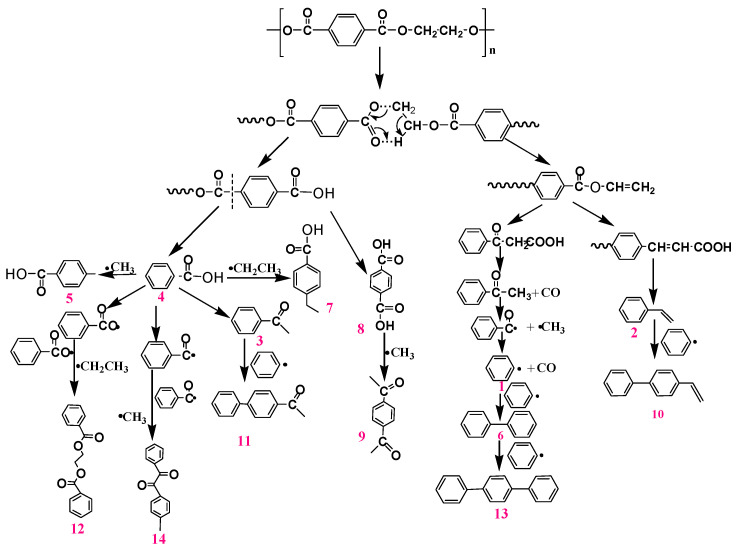
Proposed pyrolysis processes for PET.

**Figure 11 materials-14-01470-f011:**
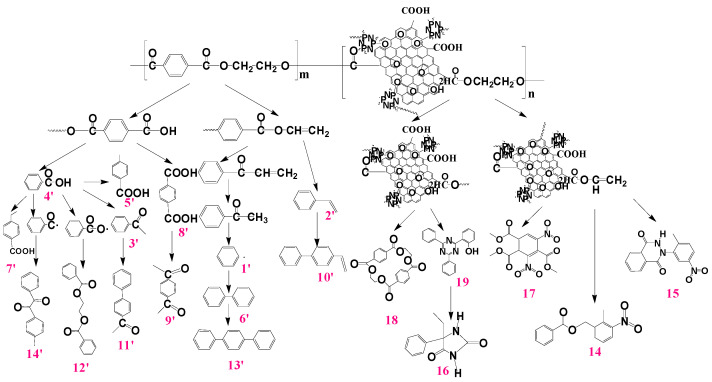
Proposed pyrolysis processes for PET-0.4G-0.2H.

**Figure 12 materials-14-01470-f012:**
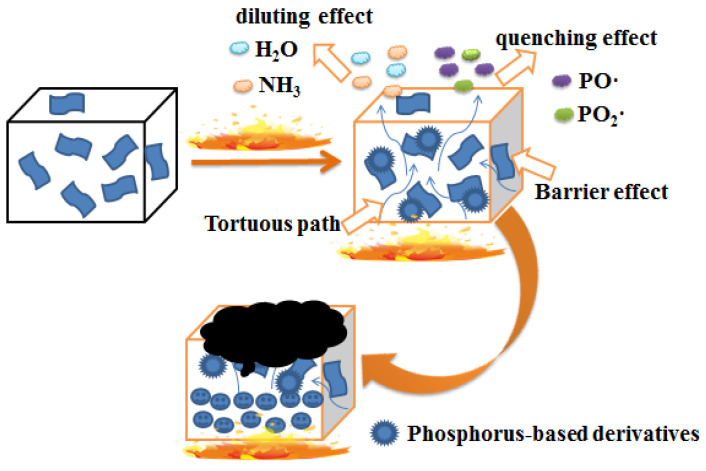
Proposed flame-retardant mechanism.

**Table 1 materials-14-01470-t001:** DSC data of neat PET and its composites.

Sample	^1^ T_g_ (°C)	^2^ T_cc_ (°C)	^3^ T_mc_ (°C)	^4^ T_m_ (°C)
PET	71.1	141.0	199.9	248.0
PET-0.4G	49.4	108.2	156.5	203.7
PET-0.2H	51.4	128.3	133.2	206.5
PET-0.4G-0.2H	72.4	143.5	199.0	250.7

^1^ T_g_ is glass transition temperature. ^2^ T_cc_ is cold crystallization temperature. ^3^ T_mc_ is melting crystallization temperature. ^4^ T_m_ is melting temperature.

**Table 2 materials-14-01470-t002:** Thermal degradation data of neat PET and its composites in N_2_.

Sample	^1^ T_5 wt%_ (°C)	^2^ T_30 wt%_ (°C)	^3^ T_50 wt%_ (°C)	^4^ T_max_ (°C)	^5^ C_w_ (wt%)
PET	381.9	412.8	425.7	428.7	8.3
PET-0.4G	383.8	411.1	422.2	425.7	8.8
PET-0.2H	391.8	420.7	432.1	435.6	6.9
PET-0.4G-0.2H	391.7	418.7	430.1	432.6	11.0

^1^ T_5 wt%_, ^2^ T_30 wt%_ and ^3^ T_50 wt%_ are the temperature at 5 wt%, 30 wt% and 50 wt% mass loss, respectively. ^4^ T_max_ is the maximum mass loss rate temperature. ^5^ C_w_ is the char residue weight at 700.

**Table 3 materials-14-01470-t003:** LOI and UL-94 of neat PET and its composites.

Samples	Composition (wt%)			LOI (%)	UL-94
	GO	HCCP (Pwt%)	GO-HCCP		
PET	0	0	0	22	V-2
^1^ PET-0.4G	0.4	0	0	21	V-2
^2^ PET-0.2H	0	0.2	0	26	V-2
^3^ PET-0.4G-0.2H	0	0	0.4-0.2	24	V-2

^1^ PET-0.4G is composed of PET and 0.4 wt% GO, ^2^ PET-0.2H is composed of PET and 0.2 Pwt% HCCP, ^3^ PET-0.4G-0.2H is composed of PET and 0.4 wt% GO and 0.2 Pwt% HCCP.

**Table 4 materials-14-01470-t004:** Cone calorimetric date of neat PET and its nanocomposites.

Sample	^1^ TTI	^2^ pHRR	^3^ THR	^4^ pSPR	^5^ COP	^6^ CO_2_P	^7^ MEHC
	(s)	(kW/m^2^)	(MJ/m^2^)	(m^2^/S)	(g/s)	(g/s)	(MJ/Kg)
PET	63 ± 2	875.9 ± 11	60.7 ± 1	0.24 ± 0.01	0.024 ± 0.02	0.62 ± 0.02	19.0 ± 0.2
PET-0.4G	50 ± 1	916.0 ± 12	71.4 ± 2	0.27 ± 0.02	0.019 ± 0.01	0.65 ± 0.02	18.9 ± 0.2
PET-0.2H	57 ± 2	618.9 ± 9	55.1 ± 1	0.21 ± 0.01	0.021 ± 0.01	0.51 ± 0.01	16.6 ± 0.1
PET-0.4G-0.2H	58 ± 1	648.6 ± 11	63.3 ± 2	0.20 ± 0.02	0.015 ± 0.01	0.43 ± 0.01	17.8 ± 0.2

^1^ TTI is the time to ignition, ^2^ pHRR and ^3^ THR are the peak of heat release rate and the total heat release, respectively, ^4^ pSPR is the peak of smoke production rate, ^5^ COP and ^6^ CO_2_P are CO production and CO_2_ production, respectively, ^7^ MEHC is the mean effective heat of combustion.

**Table 5 materials-14-01470-t005:** Quantitative comparison of flame-retardant modes for PET composites.

Sample	Barrier Effect (%)	Flame Inhibition Effect (%)	Charring Effect (%)
PET-0.4G	11.1	0	4.4
PET-0.2H	22.2	12.6	9.5
PET-0.4G-0.2H	29.1	5.6	7.6

**Table 6 materials-14-01470-t006:** Compounds identified in the pyrograms of PET and PET-0.4G-0.2H.

NO.	Retention Time, min	Name	Content, (%)	
			PET	PET-0.4G-0.2H
1, 1′	6.10	Benzene	7.6	5.2
2, 2′	7.49	Styrene	5.1	3.7
3, 3′	7.82	Acetophenone	3.9	3.1
4, 4′	7.95	Benzoic acid	34.1	39.6
5, 5′	8.53	Methyl benzoate	8.5	4.8
6, 6′	9.23	Biphenyl	7.2	4.3
7, 7′	9.47	p-ethylbenzoic acid	6.3	4.6
8, 8′	11.13	Terephthalic acid	8.7	11.3
9, 9′	12.46	p-Acetylacetophenone	5.0	7.2
10, 10′	13.56	p-Vinylbiphenyl	1.9	1.7
11, 11′	13.706	p-Phenylacetophenone	2.1	1.6
12, 12′	15.01	Ethane-1,2-diyl dibenzoate	1.7	1.4
13, 13′	15.37	Terphenyl	3.1	1.9
14, 14′	16.98	1,3-Diphenyl-1,3-propanedione	1.6	1.1
15	17.33	Phthalazine-1,4(2H,3H)-dione, 2-(2-methyl-5-nitrophenyl)	-	1.3
16	17.95	5-Ethyl-5-phenylhydantoin	-	0.9
17	20.52	1,2,4-benzenetricarboxylic acid, 3,5-dinitro-trimethyl ester	-	1.4
18	21.25	8,10,18,20,21,23-hexaene-2,7,12,17-tetrone	-	1.2
19	21.37	o-(4,6-Diphenyl-1,3,5-triazin-2-yl)phenol	-	0.8

## Data Availability

Data is contained within the article or Appendix A.

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
