# Peer review of "Functionalization of PET with Phosphazene Grafted Graphene Oxide for Synthesis, Flammability, and Mechanism"

_materials, 2021, doi:10.3390/ma14061470_

Round 1

Reviewer 1 Report

Article “Functionalization of Graphene with Phosphazene Compound for PET Composites: Synthesis, Flammability and Mechanism” deals with obtaining PET/GO-HCCP composite confirming the chemical binding and study of its combustion properties. Authors achieved slightly better results (LOI 26) than for pure PET sample (LOI 21) worse than possible on other PET-based composites (LOI 36) found in the literature.   Some corrections in the text are needed starting with the title, which is misleading. You have functionalized PET with composite GO-HCCP not reversed. Proposition “Functionalization of PET with Phosphazene grafted Graphene Oxide for Synthesis, Flammability, and Mechanism”.

Please use full names of acronyms when used for the first time, not only further in the text. Table labels should have a complete description of what is inside, not only in the text. No explanation to “0.4G”, “0.2H” “PET-0.4G, PET-0.2H”  when it first appears in text for the first time. Table 3 gives some answers to this question. Equations 1, 2, 3 missing descriptions of parameters. Missing definitions pHRR, pSPR and COP used in conclusions.

Abstract

The SEM 14 showed a better dispersion of GO-HCCP than GO in the PET matrix.  “The” is unnecessary

Introduction

When the LOI acronym is for the first time used, write the full name of it and its meaning.

Text corrections

Oxide graphene [28-32] is a two-56….. (Graphene oxide)

Zhang et al[33].found that adding FGO grafted with HCCP and 3-62 (dot)

Then 15 g EG, 0.35 g 1010 and 0.35 g 102 triphenylphosphite were added to the reaction mixture for 30 min. Do not follow 1010g? missing dot?

…of 100 mL·min-1. The samples were heated 126   (Not sure, but check if someone in this journal uses dots writing down units. You use that multiple times.)

Low quality of figures. Double figures should be resized onto two columns.

Table 1. DSC data of neat PET and its composites (Description of presented values is needed)

Table 2. Thermal degradation data of neat PET and its composites.  (Description of presented values is also needed in Table label not only in the text)

Table 4. Cone calorimetric date of neat PET and its nanocomposites. .  (Description of presented values is also needed in Table label not only in the text)

Tm decreased to lower values in PET composites. Similarly, the 213  (Tm lower index)

Author Response

Thank you for your letter and for the reviewer’ comments concerning our manuscript entitled “Functionalization of graphene with phosphazene compound for PET composites: synthesis, flammability and mechanism” (ID: 1107619). Those comments are all valuable and very helpful for revising and improving our paper, as well as the important guiding significance to our researches. We have studied comments carefully and have made correction which we hope meet with approval. Revised portion are marked in red in the paper.

The main corrections in the paper and the responds to the reviewer’s comments are as flowing:

1. Response to comment: The misleading title of the article.

Response: Considering the Reviewer’s suggestion, we have modified the title to “Functionalization of PET with Phosphazene grafted Graphene Oxide for Synthesis, Flammability, and Mechanism”.

2. Response to comment: The description and explanation of acronyms. Response: We are very sorry for our negligence of the description and explanation of acronyms, revised portion are marked in red in the paper.

3. Response to comment: The SEM 14 of the better dispersion of GO-HCCP. Response: It is really true as reviewer suggested that the necessity or unnecessity of SEM 14 However, in my opinion, the better dispersion of GO-HCCP is not only beneficial to the improvement of flame-retardant, but also can effectively improve the mechanical properties of PET. The characterization of its dispersion is more convincing for the subsequent improvement of the above performance.

4. Response to comment: The full name and meaning of LOI. Response: The full name of LOI is the limiting oxygen index, and the LOI means the lowest volume fraction concentration of oxygen in a mixture of oxygen and nitrogen which can just enough to support the combustion of polymer.

5. Response to comment: About text corrections Response: We are very sorry for our negligence of details of manuscript. We have carefully checked the text for typos; missing spaces and punctuation.

We have made correction according to the Reviewer’s comments. We tried our best to improve the manuscript and made some changes in the manuscript. These changes will not influence the content and framework of the paper. And here we did not list the changes but marked in red in revised paper.

We appreciate for Editors/Reviewers’ warm work earnestly, and hope that the correction will meet with approval. Once again, thank you very much for your comments and suggestions.

Reviewer 2 Report

The authors have presented the synthesis of a novel hybrid system based on graphite oxide and cyclotriphosphazene and applied obtained compound as a flame retardant for PET. The approach is a novelty and the presented results clearly indicate the FR properties of GO-HCCP, however, several changes in the manuscript are required prior to acceptance.

  • Introduction part:  "Hexachlorocyclotriphosphazene (HCCP)[25-27] is a typical nitrogen-phosphorus flame retardant, which hexatomic ring is composed of P and N atoms linked by alternating single and double bonds, and this unique conjugate structure endows HCCP well flame retardancy and chemical inertness" This part must be a little bit redrafted. The cyclotriphosphazene core provides high chemical and thermal stability, not HCCP (chlorine-containing), because HCCP has a tendency to ring-opening polymerization, moreover, chlorine atoms can be easily substituted by alcohols, phenols, amines. Moreover, the CP core can also survive in the presence of highly reactive compounds such as organolithium reagents (DOI: 10.1016/j.jorganchem.2017.10.030 and 10.1016/j.polymdegradstab.2018.07.026). The remarkable stability of CP core, as well as the possibility to equip CP with the functional groups through a substitution reaction with alcohols, phenols, amines, (and subsequent stability in catalytic or organometallic reactions), makes the HCCP a great starting material for functionalization, therefore, cyclophosphazene derivatives found application in flame retardancy of both, natural (cotton for example) and synthetic materials (epoxy resin, polyurethanes). The appropriate references for natural and synthetic materials modified with cyclophosphazene should be also added.
  • In the whole manuscript, the Authors use many abbreviations, however, the meaning is not provided. 
  •  Synthesis of GO-HCCP. It is hard to predict the exact structure of the target compound, therefore, the yield in % is impossible to report, however, the Authors should add a yield in grams.
  • Preparation of PET-GO-HCCP Composites: "The transesterification step finished after a theoretical amount of water was removed" The amount of water should be reported (theoretical and measured)
  • Because this is the first report describing GO-HCCP, the Authors should record a solid-state 31P NMR analysis and report the shift of phosphorous when is bonded to the GO.  
  • Figure 3a. the FT-IR spectrum of GO -HCCP revealed an appearance of the new absorption bands in the range from ~2500-2800, the new peaks come from which unit? It is not commented on in the text. In my opinion, it indicates the presence of a significant amount of impurity in the GO-HCCP, namely, triethylamine hydrochloride, because quaternary amines give signals in this range
  • Morphology:  I suggest that SEM should be supported with the energy-dispersive X-ray spectroscopy (EDS) experiments (only for GO-HCCP) in order to identify the phosphorus atom dispersion, which will clearly show the dispersion of GO-HCCP in PET matrix.
  • DSC curves are not presented in the manuscript, therefore, should be provided in supporting information (for all composites)
  • TGA analysis - the results should be also presented in a plot form. Moreover, TG analysis of prepared composites should be also performed in the air atmosphere. 
  • Line 231/232 "The result demonstrated that the addition of HCCP significantly enhanced the thermal stability and further delayed the decomposition of PET". The presence of phosphazene in PET influenced the thermal properties, however, I don't think that increase several oC is a significant change. 
  • Cone calorimetry. The results are provided in the table, however, I think that will be also good to provide experimental data as plots in supporting information. Sometimes, the shape of the curves, especially of HRR provides specific information about the material behavior when compared to the reference sample. Therefore, a reader will find it in supporting information.
  • Some graphics and plots are too small, therefore, it is very hard to read them.

Author Response

Dear Editor and Reviewer:
Thank you for your letter and for the reviewer’ comments concerning our manuscript entitled “Functionalization of graphene with phosphazene compound for PET composites: synthesis, flammability and mechanism” (ID: 1107619). Those comments are all valuable and very helpful for revising and improving our paper, as well as the important guiding significance to our researches. We have studied comments carefully and have made correction which we hope meet with approval. Revised portion are marked in red in the paper.

The main corrections in the paper and the responds to the reviewer’s comments are as flowing:
1. Response to comment: The description of the HCCP.

Response: We have rewritten this part according to the reviewer’s suggestion. Hexachlorocyclotriphosphazene (HCCP) is a typical hexatomic ring composed by P and N atoms linked by alternating single and double bonds, and this unique conjugate structure endows HCCP high chemical and thermal stability. Moreover, chlorine atoms can be easily substituted by alcohols, phenols and amines which make the HCCP a great starting material for functionalization. Therefore, cyclophosphazene derivatives found application in flame retardancy of both, natural (cotton for example) and synthetic materials (epoxy resin, polyurethanes).

2. Response to comment: The description and explanation of abbreviations.

Response: We are very sorry for our negligence of the description and explanation of abbreviations, revised portion are marked in red in the paper.

3. Response to comment: The yield of GO-HCCP.

Response: As reviewer suggested that it was hard to report the yield. Therefore, we can infer from the mass decreasing in TGA test that the grafted mass is approximately 26.62% compared with the mass of GO.

4. Response to comment: The theoretical and measured amount of water.

Response: We have made correction according to the reviewer’s comments. “The transesterification step finished after about 70 mL of water was removed.” In transesterification step, 75mL of water can be generated when the esterification ratio reaches 100%.While in practical experiment, it’s hard to achieve 100% esterification ratio, and some water will be stored in the reflux tower. So the actual water output is about 70mL.

5. Response to comment: About the solid-state 31P NMR analysis.

Response: It is really true as reviewer suggested that the solid-state 31P NMR analysis of GO-HCCP. However, FTIR, and XPS results can certify the successful grafting onto the surface of GO which showed the increasing layers pacing of GO in XRD and the disordering structure in Raman. So in my opinion, the solid-state 31P NMR analysis of GO-HCCP is unnecessary.

6. Response to comment: About the new absorption bands in FT-IR spectrum.

Response: It is really true as reviewer suggested that the absorption bands in the range from ~2500-2800 cm-1 corresponded to triethylamine hydrochloride which indicates the products should be further purification.

7. Response to comment: EDS experiments for GO-HCCP

Response: Considering the reviewer’s suggestion, we have provided the EDS experiments for GO-HCCP and the result have been presented in the article.

 8. Response to comment: DSC curves

Response: Considering the reviewer’s suggestion, we have provided the DSC curves in supporting information.

9. Response to comment: TGA curves

Response: Considering the reviewer’s suggestion, we have provided the TGA curves and the TG analysis of prepared composites in the air atmosphere in supporting information.

10. Response to comment: About the word of “significantly”

Response: We are very sorry for our negligence of the rigor of description. These increasing temperature are not significant, and these parameters are not sufficient to justify a significant effect. We will remove such loose words from the article.

11. Response to comment: Cone curves

Response: Considering the reviewer’s suggestion, we have provided the cone curves in supporting information.

12. Response to comment: The size of graphics and plots.

Response: We are very sorry for our negligence of the size of graphics and plots. We have provided graphics and plots of moderate size and clarity.

We tried our best to improve the manuscript and made some changes in the manuscript. These changes will not influence the content and framework of the paper. And here we did not list the changes but marked in red in revised paper. We appreciate for Editors/Reviewers’ warm work earnestly, and hope that the correction will meet with approval.
Once again, thank you very much for your comments and suggestions.

Reviewer 3 Report

This paper concerns the study of a PET composite containing a novel flame-retardant based on Graphene Oxide functionalized with phosphazene compound. The synthesis of the materials has been described, a comprehensive characterization has been performed by several analytical techniques. Finally, flammability has been investigated and a possible flame-retardant mechanism has been proposed.

I suggest only a few minor changes before the publication.

- Please, specify the acronyms at their first appearance (especially those reported in the Introduction).

- Section 2.4: report details about the ATR crystal used to acquire the FTIR spectra.

- Lines 143-151: report the standards' details in the reference list.

- Table 1: add a legend to explain the meaning of the reported data.

- Line 209: cite a few references to support this statement.

- The acronyms used in Equations 1-3 make the formulas difficult to follow; try to simplify them and/or specify each term used in these equations.

- Figure 5: I would suggest removing the lines; in my opinion, these data cannot directly be related to each other.

- Lines 392-404: please, try to better highlight differences and/or similarities to the already known flame-retardant mechanisms.

- Figure 2, 3, 6, 7, 8, 9, and 10 are difficult to read; please, increase the size.

- Carefully check the manuscript; there are many typos, in particular, missing spaces and punctuation.

Author Response

Dear Editor and Reviewer:
Thank you for your letter and for the reviewer’ comments concerning our manuscript entitled “Functionalization of graphene with phosphazene compound for PET composites: synthesis, flammability and mechanism” (ID: 1107619). Those comments are all valuable and very helpful for revising and improving our paper, as well as the important guiding significance to our researches. We have studied comments carefully and have made correction which we hope meet with approval. Revised portion are marked in red in the paper.

The main corrections in the paper and the responds to the reviewer’s comments are as flowing:

1. Response to comment: The description and explanation of acronyms.

Response: We are very sorry for our negligence of the description and explanation of acronyms, revised portion are marked in red in the paper.

2. Response to comment: About the ATR crystal used to acquire the FTIR spectra.

Response: The ATR crystal used to acquire the FTIR spectra is diamond. The spectra were collected in the optical range of 400-4000 cm-1 with a scanning number of 64.

3. Response to comment: The standards’ detail

Response: As reviewer suggested that the article has listed the standards and the size of sample. Details of standard are available online.

4. Response to comment: The legend to explain the meaning of the reported data

Response: We are very sorry for our negligence of the legend that explaining the meaning of the reported data. We will provide the detailed explain under the table.

5. Response to comment: Cite references to support statement

Response: Considering the reviewer’s suggestion, we have cited some references to support the statement. The additional references are marked in red in the paper.

6. Response to comment: Equations 1-3

Response: We are very sorry for our negligence of confusing the subscript and case. The parameters in the equation are all cone data, and we have provided detailed comments below the equation.

7. Response to comment: About Figure 5

Response: Considering the Reviewer’s suggestion, we have changed the diagram into histogram. The data is irrelevant, so it cannot be represented by a line chart.

8. Response to comment: Highlight differences and/or similarities

Response: We have re-written this part according to the Reviewer’s suggestion. (â…°) the “tortuous path effect” and “barrier effect” of GO can insulate the permeation of oxygen and volatile flammable gases, thereby decreasing the release of heat and smoke. (â…±) the diluting and quenching effect of HCCP can reduce the concentration of oxygen and flammable gases, which can restrain combustion in gas phase effectively. (â…²) phosphorus-based derivatives formed in condensed phase can significantly promote the formation of protective layer, preventing composites from the radiant heat and direct contact with flame.

9. Response to comment: The size of figure.

Response: We are very sorry for our negligence of the size of figure. We have provided figure of moderate size and clarity.

10. Response to comment: The writing error in the manuscript

Response: We are very sorry for our negligence of details of manuscript. We have carefully checked the text for typos, missing spaces and punctuation.

We tried our best to improve the manuscript and made some changes in the manuscript. These changes will not influence the content and framework of the paper. And here we did not list the changes but marked in red in revised paper. We appreciate for Editors/Reviewers’ warm work earnestly, and hope that the correction will meet with approval.
Once again, thank you very much for your comments and suggestions.

Round 2

Reviewer 2 Report

The authors have successfully improved the manuscript and the presented results for sure deserve to be published in Materials. However, I cannot agree with the one statement that still exists in the Introduction section. "Hexachlorocyclotriphosphazene (HCCP) is a typical hexatomic ring composed by P and N atoms which is connected by single and double bonds, and this unique conjugate structure endows HCCP high chemical and thermal stability." not HCCP but its derivatives

In my first review, I underlined that HCCP is not a thermally and chemically stable compound in contrast to its derivatives (modified hccp, without chlorine atoms). Please see your TG curves, pure HCCP decomposed in the temperature below 200oC, therefore HCCP cannot be considered as a highly stable compound. Moreover, if the HCCP can be easily functionalized with alcohols, phenols, and amines through substitution reaction, that means HCCP is not chemically stable, but HCCP derivatives for sure are. In the previous revision, I suggested two papers that clearly corroborate the chemical stability. Moreover, when I suggested that a small part should be rewritten I didn't expect that the Authors will copy the fragment from my review and paste it into the manuscript, it was just a suggestion of what should be added to the introduction.  Therefore, I hoped that the Authors will rewrite it in their own way. If authors don't want to change the fragment presented below, it's okay, however, the appropriate references to each type of material should be added as suggested in my previous revision (for polyurethanes, epoxy resin, cotton)

"Moreover, chlorine atoms can be easily substituted by alcohols, phenols and amines which makes the HCCP a great starting material for functionalization. Therefore, cyclophosphazene derivatives found application in flame retardancy of both, natural (cotton for example) and synthetic materials (epoxy resin, polyurethanes)." 

Author Response

Dear Editor and Reviewer:
Thank you for your letter and for the reviewer’ comments concerning our manuscript entitled “Functionalization of PET with Phosphazene grafted Graphene Oxide for Synthesis, Flammability, and Mechanism” (ID: 1107619). Those comments are all valuable and very helpful for revising and improving our paper, as well as the important guiding significance to our researches. We have studied comments carefully and have made correction which we hope meet with approval. Revised portion are marked in red in the paper. The main corrections in the paper and the responds to the reviewer’s comments are as flowing:

Response to comment: The description of the HCCP.

Response: We have rewritten this part according to the reviewer’s suggestion. Hexachlorocyclotriphosphazene (HCCP) is a typical hexatomic ring composed by P and N atoms which connected by single single and double bonds. This structure with high phosphorus and nitrogen content endows HCCP great potential in flame retardancy. In addition, the chlorine atoms can be easily substituted by alcohols, phenols and amines which not only make HCCP a great starting material for functionalization, but also enhance the thermal and chemical stability of HCCP. Therefore, cyclophosphazene derivatives found application in flame retardancy of both, natural (cotton) and synthetic materials (epoxy resin, polyurethanes).

We tried our best to improve the manuscript and made some changes in the manuscript. These changes will not influence the content and framework of the paper. And here we did not list the changes but marked in red in revised paper. We appreciate for Editors/Reviewers’ warm work earnestly, and hope that the correction will meet with approval.
Once again, thank you very much for your comments and suggestions.
